# Pasta as a Source of Minerals in the Diets of Poles; Effect of Culinary Processing of Pasta on the Content of Minerals

**DOI:** 10.3390/foods10092131

**Published:** 2021-09-09

**Authors:** Karolina Jachimowicz, Anna Winiarska-Mieczan, Ewa Baranowska-Wójcik, Maciej Bąkowski

**Affiliations:** 1Department of Bromatology and Food Physiology, University of Life Sciences in Lublin, Akademicka St. 13, 20-950 Lublin, Poland; anna.mieczan@up.lublin.pl; 2Department of Biotechnology, Microbiology and Human Nutrition, University of Life Sciences in Lublin, Skromna St. 8, 20-704 Lublin, Poland; ewa.baranowska@up.lublin.pl; 3Institute of Animal Nutrition and Bromatology, University of Life Sciences in Lublin, Akademicka St. 13, 20-950 Lublin, Poland; maciej.bakowski@up.lublin.pl

**Keywords:** pasta, cereal products, minerals, culinary processing, food technology, salt, sodium intake, rinsing, minerals retention, nutrition

## Abstract

Pasta is a product that requires culinary processing which can affect the content of minerals in the finished product. The study aimed to examine how cooking pasta (1) in salted water (1 teaspoon—16 g of salt per 1 litre of water) or unsalted water and (2) rinsing cooked pasta with running water affects the content of minerals. Thirty-five samples of six types of pasta were analysed. The content of minerals was determined using the ICP-OES method. Retention of minerals in the cooked pasta was calculated. Taking the culinary treatment into account, the intake of minerals with pasta was assessed for children, adolescents, and adults, and the values were compared with the recommendations for the population of Poland. The analysed culinary factors had a statistically significant influence on the content of minerals. Adding salt to water when cooking pasta significantly increased the content of Na in the product, which in turn was negatively correlated with the content of other minerals. When pasta was cooked in unsalted water, it contained less Na and more other minerals than pasta cooked in salted water. Rinsing of pasta reduced the content of all minerals. Pasta is an important source of Mg, Cu, and Mn in the diet of Poles. These ingredients are particularly important to ensure correct development and functioning of the human body. The best method of culinary treatment of pasta is cooking in unsalted water without rinsing.

## 1. Introduction

Cereal products are the basic food for people all over the world. In 2019, the average monthly consumption of cereal products in Poland amounted to 4.23 kg, where nearly 10% was pasta [1]. Due to their intake volume, cereals are the primary source of minerals in the human diet. Our previously published studies showed that in the population of Poland, cereals cover the requirement of an adult person for K and Mg in ca. 10–15%, Na in nearly 49%, Cu and Mn in more than 55%, with Fe and Zn in ca. 20–30% [2,3,4]. The most often consumed product is bread, and pasta comes second [1].

Pasta is common in the diets of both Poles and inhabitants of other countries. Due to its neutral taste, it can be served with meat dishes and added to salads and desserts [5,6]. Consumers on the Polish market are offered a wide range of pasta. It differs in shape, dimensions, and ingredients. The basic raw materials for making pasta are semolina and/or pasta flour and water [7]. According to Statistical Yearbooks, the average monthly consumption of pasta per capita in one household in 2019 was 0.4 kg [1]. This is more than the consumption of groats (0.10 kg) and rice (0.15 kg), but less than bread (2.98 kg). Pasta is a source of fibre, carbohydrates, protein, vitamins, and macro- and microelements. The nutritional value of pasta is enriched with various additives: eggs, protein concentrates, milk, yeasts, vegetables, algae, and soybean meal.

The diet of Poles is deficient in most minerals but contains excessive amounts of Na [2,3,4]. A similar relationship can also be observed in other countries [8,9,10]. Due to the consumption volume, pasta can be an important source of deficit minerals in the diet. However, the most controversial element is Na, the consumption of which should be reduced. Sodium is necessary for living because it regulates the water and electrolyte management, maintains acid and alkaline balance, and ensures correct transmission of nerve impulses and correct functioning of muscle cells [11]. Large amounts of sodium can have serious consequences to health, including an increased risk of hypertension and cardiovascular diseases [12]. The predominant sources of sodium in a diet are processed foods and salt added to meals [13,14]. The content of sodium in meals increases when salt is added in the course of processing and cooking [15]. The content of salt in packed dry pasta is insignificant (<0.01 g/100 g), but when it is added while cooking, its level may significantly increase. The NHANES study in the United States showed that 11% of sodium comes from salt added to meals [16]. Pasta is a product salted during thermal processing. It is one of the top 12 foodstuffs contributing to excessive consumption of sodium [17]. In the NHANES study, 40% of respondents indicated that they very often used salt in cooking. In addition, the authors of the study found that pasta is one of the five most popular dishes to which salt is added [16]. The producers of pasta available in Poland recommend cooking pasta in water with salt added; the packaging gives imprecise information that water should be salted “to taste”.

The parameters recommended for cooking dry pasta differ depending on the producers’ indications on packaging and the habits of consumers. The end properties of pasta can be affected by the water-to-pasta ratio, cooking time, cooking, rinsing of the cooked pasta, amount of salt added, and other factors which can differ between the recommended and customary practices [18]. Determining the effect of various cooking parameters would make it possible to estimate the content of sodium and minerals in cooked pasta consumed by a given population, which—in turn—could be directly reflected in recommendations for cooking pasta. Therefore, this work aimed to examine the effects of cooking pasta in salted and unsalted water on the content of macroelements (Na, K, Ca, Mg) and microelements (Zn, Cu, Fe, Mn). In Poland, after cooking, pasta is customarily rinsed with running water to prevent it from sticking together. Therefore, in addition, the effects of rinsing pasta on the content of minerals in the product was examined. This paper is part of a project aiming to estimate the intake of minerals (both toxic and essential) in the Polish population.

## 2. Materials and Methods

### 2.1. Study Material

Thirty-five samples of pasta of different types and with different ingredients were analysed (Table 1 and Appendix A). In none of the products did the producer declare added salt. The products were purchased in August 2019 in grocery stores in Lublin (eastern Poland). All the products were before their expiry date, and 100 g of pasta was weighed from each sample (with the accuracy of 1 decimal place). It was cooked *al dente* in water (100 g of pasta per 1 L of drinking water) with or without table salt (1 flat tablespoon of salt per 1 L of drinking water, which corresponds to ca. 16 g of salt and 6 g of Na), according to the scheme:
NaCl + NR—cooked with salt and not rinsed;NaCl + R—cooked with salt and rinsed;NR—cooked without salt and not rinsed;R—cooked without salt and rinsed.


The cooking time was consistent with the recommendations of the pasta producers (Table 1 and Appendix A). After cooking, the pasta was strained in a plastic strainer. Next, about half of each pasta sample was rinsed with cold running water for 10 s. The cooked rinsed and non-rinsed pasta was weighed again after two minutes. The prepared pasta was placed in a drier at a temp. of 65 °C for 24 h. Afterwards, it was ground in an electric grinder fitted with plastic cutters. The ground samples were put one by one in separate, tightly closed plastic containers that were stored for one day until further chemical analyses at room temperature.

### 2.2. Chemical Analyses

After manual mixing of material from each sample, 3 g from an averaged collective sample were weighed in three replications. The control sample and the study sample were prepared in an identical manner. They were mineralised in a muffle furnace (temp. 550 °C, time 12 h, antioxidant—ydrogen peroxide), as described elsewhere [19]. The ash obtained as a result of mineralisation was dissolved in 10 mL of 1M HNO_3_ [20]. The content of Na, K, Ca, Mg, Zn, Cu, Fe, and Mn was determined by means of ICP-OES (Inductively Coupled Plasma—Optical Emission Spectrometers) using a 720-ES spectrometer (Varian, Palo Alto, CA, USA). ICP-OES operating conditions: RF generator power—1.2 kW; plasma gas flow rate—15.0 dm^3^ min^−1^; auxiliary gas flow rate—2.25 dm^3^ min^−1^; nebulizer gas flow rate—0.70 dm^3^ min^−1^; sample flow rate—0.1 mL min^−1^; replications—4; read time—15 s; peristaltic pump rotation—12 rpm. Determination parameters:
Na: wave length 589.592 nm, LOD (limit of detection) 0.25 mg kg^−1^, LOQ (limit of quantification) 0.39 mg kg^−1^;K: wave length 769.897 nm, LOD 1.04 mg kg^−1^, LOQ 1.27 mg kg^−1^;Ca: wave length 422.673 nm, LOD 0.03 mg kg^−1^, LOQ 0.05 mg kg^−1^;Mg: wave length 280.270 nm, LOD 0.03 mg kg^−1^, LOQ 0.08 mg kg^−1^;Zn: wave length 202.548 nm, LOD 0.06 mg kg^−1^, LOQ 0.09 mg kg^−1^;Cu: wave length 213.598 nm, LOD 0.03 mg kg^−1^, LOQ 0.06 mg kg^−1^;Fe: wave length 238.204 nm, LOD 0.03 mg kg^−1^, LOQ 0.05 mg kg^−1^;Mn: wave length 259.372 nm, LOD 0.10 mg kg^−1^, LOQ 0.23 mg kg^−1^.


The limits of detection (LOD) were calculated using the formula CL = 3 Sb/m [21], where “Sb” is the standard deviation of six replicate blank measurements and “m” is the slope value in the calibration curve.

In order to draw the calibration curve, LCG standards were used for the preparation of mineral solutions with concentrations of 1, 2, 4, and 8 µg per 1 L. The correctness of results was validated by means of the 1M nitric acid (HNO_3_) and two reference samples: LGC 7173 Poultry Feed and NCS ZC 73,009 Wheat. The rate of recovery of the analysed minerals ranged from 94 to 107% (Table 2). All the analyses were performed in three replications.

### 2.3. Reagents and Reference Materials

The solutions were prepared using deionized water (deionizer Hydrolab, Gdańsk, Poland) and ultra-pure chemical reagents. Hydrogen peroxide (30% pure) and nitric acid (65% ultra-pure) were purchased from POCH S.A. (Lublin, Poland). The standard solutions of Ultra Scientific minerals (1000 mg L-1, 99.99% purity) was purchased from LGC Standards Sp. z o.o. (Kiełpin, Poland). The certified reference material (CRM) was used to check the quality:(1)LGC 7173 Poultry feed (LGC Standards GmbH, Wesel, Germany) contained 1180 mg Na, 7480 mg K, 17,800 mg Ca, 2037 mg Mg, 78 mg Zn, 14 mg Cu, 148 mg Fe and 90 mg Mn per 1 kg;(2)NCS ZC 73,009 Wheat (National Institute of Standard and Technology, Gaithersburg, MD, USA) contained 17.0 mg Na, 1400 mg K, 340 mg Ca, 450 mg Mg, 11.6 mg Zn, 2.71 mg Cu, 18.5 mg Fe and 5.4 mg Mn per 1 kg.

### 2.4. Calculations and Statistical Analysis

The rate (%) of retention of minerals was calculated from the formula [22,23]:$$\mathrm{Retention}\;\left(\%\right)\;=\frac{\mathrm{mineral}\;\mathrm{content}\;\left(g\right)\;\mathrm{in}\;\mathrm{cooked}\;\mathrm{pasta}\times \mathrm{weight}\;\left(g\right)\;\mathrm{of}\;\mathrm{pasta}\;\mathrm{after}\;\mathrm{cooking}}{\mathrm{mineral}\;\mathrm{content}\;\left(g\right)\;\mathrm{in}\;\mathrm{raw}\;\mathrm{pasta}\times \mathrm{weight}\;\left(g\right)\;\mathrm{of}\;\mathrm{pasta}\;\mathrm{before}\;\mathrm{cooking}}\times 100$$

Based on the mean content of Na, K, Ca, Mg, Zn, Cu, Fe, and Mn in cooked pasta, the content of minerals was calculated per 1 serving of pasta, assuming that the serving is 140 g for adults [24], 50 g for children aged 1–3 [25], 60 g for children aged 4–6 [26], 100 g for children aged 7–12 [27], 125 g for children aged 13–15 [28], and 135 g for adolescents aged 16–18 [28]. The mean share of pasta in the reference daily intake was calculated from the formula:$$\mathrm{Share}\;\mathrm{in}\;\mathrm{reference}\;\mathrm{daily}\;\mathrm{intake}=\frac{\mathrm{Daily}\;\mathrm{dietary}\;\mathrm{intake}\times 100}{\mathrm{Reference}\;\mathrm{daily}\;\mathrm{intake}}$$
where:Daily dietary intake=Daily portion size×average content of minerals in pasta.

The intake of minerals with pasta was compared against Polish RDA—Recommended Dietary Allowances (Ca, Mg, Cu, Fe, Zn) or AI—Adequate Intake (Na, K, Mn) [29]. The RDA and AI values for minerals according to Polish norms are presented in Appendix A.

The calculations were made by using Statistica 13.1 software. Mean values were calculated based on three replications per sample. Statistically significant differences (*p* < 0.05) were computed by single-factor analysis of variance (ANOVA), using Duncan’s test. Correlations between the minerals evaluated were obtained using Pearson’s correlation coefficient (r).

## 3. Results

### 3.1. How the Type of Culinary Treatment Affects the Content of Minerals in Pasta

A statistically significant influence of salt added to water, rinsing the cooked pasta with running water and the type of pasta on the content of minerals was found. Appendix A contain detailed information.

In NaCl + NR pasta, a strong positive correlation was noted between the content of Mn and K (r = 0.827). On the other hand, NaCl + R pasta showed a strong positive correlation between the content of Zn and the content of K and Mg (r > 0.9), and a strong negative correlation between the content of Mg and Fe (r = −0.882). The relationship between the content of the analysed minerals can be presented as Mg > K > Na > Ca > Mn > Fe > Zn > Cu for NaCl + NR and Mg > K > Na > Ca > Fe > Mn > Cu > Zn for NaCl + R. In NaCl + NR, the retention of Na was, on average, 780%, and in NaCl + R pasta, ca. 300% (Table 3). In NaCl + NR pasta, the retention of K was 254%, and that of Zn, ca. 70%, whilst retention of other analysed minerals ranged from 116 to 163%. In NaCl + R pasta, the retention of K was ca. 190%, Fe—132%, and Zn—32%. The retention of Ca, Mg, Cu, and Mn was close to 100%. Pasta cooked in unsalted water (NR, R) contained less (*p* < 0.05) minerals than raw pasta.

The relationship between the content of analysed minerals can be presented as Mg > K > Ca > Na > Mn > Fe > Zn > Cu for NR pasta and Mg > K > Ca > Na > Mn > Fe > Zn > Cu for R pasta. Retention of minerals is presented in Table 3.

At the same time, some general trends in the content of respective minerals can be observed depending on the type of culinary treatment. Irrespective of the type of pasta, cooking in salted water (NaCl + NR, NaCl + R) increased (*p* < 0.05) the content of Na (Figure 1a) and decreased (*p* < 0.05) the content of all other minerals in comparison to their level in raw pasta (Figure 1b–d; Figure 2a–d). In turn, pasta cooked in unsalted water (NR, R) contained less (*p* < 0.05) analysed minerals (except K) than raw pasta, irrespective of its type (Figure 1 and Figure 2). Furthermore, cooking pasta in unsalted water (NR, R) led to a decreased (*p* < 0.05) content of Na and an increased level (*p* < 0.05) of other minerals than in pasta cooked in salted water (NaCl + NR, NaCl + R).

Rinsing pasta cooked in salted water (NaCl + R) decreased the content of Zn and Mn by about 50%, and that of other minerals by 20–30% compared to NaCl + NR (Table 4). Rinsing pasta cooked in unsalted water (R) decreased the content of Na by ca. 8%, the content of K, Cu, Fe and Mn by 14–18%, and that of Ca, Mg and Zn by 25–29% in comparison to non-rinsed pasta (NR). Salt added to water while cooking pasta increased the content of Na nearly 3.5 times for rinsed pasta and nearly 5 times for non-rinsed pasta compared to pasta cooked in unsalted water. Rinsing of pasta cooked in salted water (NaCl + R) led to a decrease in the content of Zn by more than 70%, of Mn by more than 50%, of K, Mg and Cu by ca. 40%, and of Fe and Cu by less than 30% as compared to R pasta. On the other hand, the NaCl + NR pattern decreased the content of Zn by ca. 60%, of Mg by more than 40% and that of other elements by a maximum of 30% compared to NR pasta.

### 3.2. Pasta as a Source of Minerals in the Diets of Poles

#### 3.2.1. Sodium, Potassium, Calcium, Magnesium

The supply of Na with one serving of pasta cooked in salted water (NaCl + NR, NaCl + R) is higher (*p* < 0.05) than with one serving of pasta cooked in unsalted water (NR, R) in all consumer age groups (Figure 3). This relationship can be presented as NaCl + NR > NaCl + R > NR = R. In the diet of adults, one serving of pasta will cover the requirement for Na from 0.1% AI (R) to 0.8% AI (NaCl + NR), in the diet of children aged 1 to 6—from 0.1% AI (NR, R) to 0.5% AI (NaCl + NR), in the diet of children aged 7 to 12—from 0.1% AI (NR, R) to 0.6% AI NaCl + NR), and in that of adolescents aged 13 to 18—from 0.1% AI (NR, R) to 0.7% AI (NaCl + NR) (Table 5).

The intake of K with one serving of cooked pasta can be presented as NR > R > NaCl + NR > NaCl + R (Figure 3). Coverage of the daily requirement of adults and adolescents aged 16–18 for K with one serving of pasta ranges from 0.3% AI (NaCl + R) to 0.6% AI (NR), in the diet of children aged 1 to 6—from 0.5% AI (NaCl + R) to ca. 0.9% AI (NR), children aged 7 to 12—from 0.4% AI (NaCl + R) to 0.7% AI (Table 5).

In the diet of adults, one serving of cooked pasta will lead to the intake of Ca in the amount covering from 0.3% RDA (NaCl + R) to ca. 0.5% RDA (NR), in the diet of children aged 1 to 3—from 0.1% RDA (NaCl + R, R) to 0.3% RDA (NaCl + NR, NR), in the diet of children aged 4 to 6—from 0.1% RDA (NaCl + R) to 0.2% RDA (NaCl + NR, NR, R), and in the diet of older children and adolescents—from 0.2% RDA (NaCl + R) to 0.4% RDA (NR) (Table 5).

The supply of Mg with pasta can be presented as NR > R > NaCl + NR > NaCl + R (Figure 3). In Polish norms, the requirement of women and men for Mg differs—one serving of cooked pasta supplies from 11% RDA (NaCl + R) to 25% RDA (NR) of Mg to women, and from 8.5% RDA (NaCl + R) to 19% RDA (NR) (Table 6) to men. In the diet of children aged 1–3, a serving of pasta supplies from 15% RDA (NaCl + R) to 35% RDA (NR) of Mg, in the diet of children aged 4 to 6—from 11.5% RDA (NaCl + R) to 26.2% RDA (NR), and in that of children aged 7 to 12—from 13.5% RDA (NaCl + R) to ca. 30% RDA (NR). According to Polish norms, the requirement for Mg in the case of adolescents aged over 13 depends on the sex. Girls will receive from ca. 9% RDA (NaCl + R) to ca. 20% RDA (NR) of Mg with one serving of pasta, and boys—from ca. 8% RDA (NaCl + R) to 18% RDA (NR) (Table 5).

#### 3.2.2. Zinc, Copper, Iron, Manganese

The supply of Zn, Cu, Fe, and Mn with a single serving of cooked pasta can be represented as NR > R > NaCl + NR > NaCl + R (Figure 4). In Polish norms, the requirement of women and men for Zn differs—one serving of cooked pasta supplies from 0.6% RDA (NaCl + R) to 3% RDA (NR) of Zn to women, and from 0.4% RDA (NaCl + R) to 2% RDA (NR) (Table 6) to men. In the diet of children aged 1–3, a serving of pasta supplies from 0.5% RDA (NaCl + R) to nearly 3% RDA (NR) of Zn, of children aged 4 to 6—from 0.4% RDA (NaCl + R) to 2% RDA (NR), and in that of children aged over 7 and adolescents—from 0.5% RDA (NaCl + R) to ca. 2.5% RDA (NR).

In the diet of adults and adolescents aged 16–18, one serving of cooked pasta supplies from 6.6% RDA (NaCl + R) to ca. 12.5% RDA (NR) of Cu, in the diet of children aged 1 to 3—from 7% RDA (NaCl + R) to ca. 14% RDA (NR), and in that of children aged 7 to 15—from ca. 6% RDA (NaCl + R) to ca. 11.5% RDA (NR) (Table 6).

One serving of pasta in the diet of adult women supplies from ca. 1% RDA (NaCl + R) to ca. 1.6% RDA (NR) of Fe, and in that of men—from ca. 1.7% RDA (NaCl + R) to 2.9% RDA (NR) (Table 6). In the diet of children aged 1–3, one serving of cooked pasta supplies from 0.9% AI (NaCl + R) to 1.5% AI (NR) of Fe, in the diet of children aged 4 to 6—from 0.7% AI (NaCl + R) to 1.3% AI (NR), and in that of children aged over 7—from 1.2% AI (NaCl + R) to ca. 2.1% AI (NR). In Polish norms, the requirement for Fe in the case of adolescents aged over 13 depends on the sex. Girls will receive from ca. 1% RDA (NaCl + R) to ca. 2% RDA (NR) of Fe with one serving of pasta, and boys—from ca. 1.3% RDA (NaCl + R) to ca. 2.2% RDA (NR).

In Polish norms, the requirement for Mn in the case of adolescents aged over 13 and adults also depends on the sex. One serving of cooked pasta will supply from 7.1% AI (NaCl + R) to 18.3% AI (NR) of Mn to adult women, and from 5.6% AI (NaCl + R) to 14.3% AI (NR) (Table 6) to men. Girls aged 13–15 will receive from 7.1% AI (NaCl + R) to 18.3% AI (NR) of Mn with one serving of pasta, and boys—from 5.0% AI (NaCl + R) to 13.3% AI (NR). In turn, adolescents aged 16–18 will receive from ca. 8% AI (NaCl + R) to ca. 20% AI (NR) of Mn (women) with one serving of pasta, and from ca. 6% AI (NaCl + R) to more than 14% AI (NR) (men). In the diet of children aged 1–6, one serving of pasta will supply from ca. 4% AI (NaCl + R) to ca. 10% AI (NR) of Mn, and in that of children aged 7 to 12—from 5.4% AI (NaCl + R) to ca. 14% AI (NR).

Table 7 summarizes relationships between the content of minerals in the analysed pasta for each cooking method.

## 4. Discussion

### 4.1. How Culinary Processes Affect the Content of Minerals in Pasta

The process of cooking cereal products modifies their chemical composition as a result of the removal or absorption, and in particular, the content, of minerals that are easily soluble in water [23,30]. In the presented study, cooking pasta in salted water increased the content of Na and decreased that of all other analysed minerals in comparison to raw pasta. In turn, pasta cooked in unsalted water contained less analysed minerals (all except K) than raw pasta. Furthermore, cooking pasta in unsalted water led to a decreased content of Na and other minerals in comparison to pasta cooked in salted water. Due to increasing the content of Na in water by adding salt, Na from the solution penetrates into the product, thereby reducing the content of other minerals, while cooking in unsalted water causes Na to penetrate from the product into the water. The degree of penetration of minerals from pasta into water and the other way round during cooking depends on the content of Na in the water [24]. The studies by Albrecht et al. [23] showed that cooking pasta in salted water (from 4.4 to ca. 6.0 g NaCl per 1 litre) increased the concentration of Na in pasta about 100 times in comparison to its concentration in pasta cooked in unsalted water, whereas cooking in unsalted water reduced the content of Na by 16–69% in comparison to raw pasta. In turn, rinsing of cooked pasta with running water decreased the content of Na by ca. 30% [23]. The quoted study also demonstrated K > Mg > Ca > Fe > Zn > Mn > Cu in cooked pasta, irrespective of whether it was cooked in water with or without salt. In the presented study, the addition of salt increased the content of Na at the expense of Ca, while rinsing pasta cooked in salted water changed the relationship between the microelements: the content of Fe was higher than that of Mn, and that of Cu was higher than Zn, while the use of other experimental factors resulted in Mn > Fe > Zn > Cu. According to Cubadda et al. [31], cooked Italian pasta contained Fe > Zn > Mn > Cu. Bianchi et al. [24] added different amounts of salt to water used for cooking pasta: 0 g, 3.17 g or 6.34 g of salt per litre, which led to a linear increase in the content of Na in the pasta. In this study, using water without salt reduced the content of Na in cooked pasta by 48%, in comparison to the content in pasta cooked in water with 3.17 g of salt and by 99% in comparison to the content in pasta cooked in water with 6.34 g of salt. Similarly, Jambrec et al. [32] demonstrated that when pasta is cooked in unsalted water, the content of Na in the product is reduced by about 42% in comparison to the raw product.

According to Manthey and Hall [30], cooking pasta enriched with buckwheat meal in distilled water reduces the content of K, Cu, and Zn by 62%, 45%, and 11%, respectively. In the opinion of these authors, cooking generally had no influence on the content of Fe and Mg, while the content of Ca and Mn after cooking was similar or higher than prior to cooking. According to Jambrec et al. [33], in the course of cooking, the pasta loses ca. 50% K. The losses of Ca and Mg were not significant; the content of Ca in cooked pasta was even higher than in raw pasta, which, according to the authors, can be attributed to a change in the proportions of ingredients (flushing of starch and protein). Our own study showed that the content of Fe, Mg, Ca, and Mn after cooking was lower than in raw pasta, both in rinsed and non-rinsed, and in salted and unsalted samples. The unchanged content of certain minerals does not mean that they were not flushed from pasta during cooking, but rather, were leached in the proportion similar to other minerals [30]. Other reference sources suggest that the decrease in the content of minerals by 10–30% during traditional cooking of pasta was typical, except for K of which 67% were retained [34]. The study by Albrecht et al. [23] noted that the content of K and Cu decreased by 58% and 88%, respectively, when pasta was traditionally cooked in salted water. Similarly, Yaseen [34] and Ranhotra [22] reported that about 67% of K is flushed from traditionally cooked pasta. In our own study, the highest loss of K was recorded for pasta cooked in salted water and then rinsed (NaCl + R)—ca. 37% in comparison with raw pasta. Ikeda and Shimizu [35] observed that ca. 20% of Zn is flushed from buckwheat pasta in cooking. In our own study, the loss of this element in cooking reached up to 89% for pasta cooked in salted water and then rinsed (NaCl + R). Cubadda et al. [31] found an increased content of Ca in pasta cooked in drinking water, both salted and unsalted. When pasta is cooked in distilled water, the increase of Ca can be explained by the flushing of starch and soluble proteins, which alters the direct composition of the cooked pasta [36].

In the presented studies, rinsing of the cooked pasta decreased the content of all the analysed minerals. The differences were particularly high for Na, the content of which decreased by up to 31% for pasta cooked in salted water. Other authors obtained similar results, finding that rinsing of the cooked pasta with running water reduces the content of Na by about 30% [23,24]. According to Ranhotra et al. [22], the retention of Na while cooking pasta in unsalted distilled water can be from 7% up to 89%, for other minerals (K, Ca, Mg)—from 25% to 114%, and for microelements (Zn, Cu, Fe, Mn)—from 62% to 119%. Retention of some elements was higher than 100% in the quoted study. This can be due to the fact that apparent retention, not taking into account the loss of solids in cooking, often shows falsely high values [37]. Another reason may be the highly mineralized tap water in Lublin [38], from which the minerals were transferred to the cooked product. It is possible that hard tap water may increase the retention of minerals in prepared meals. Another reason may be changes in the proportion of minerals caused by the transfer of minerals from water to pasta and from being rinsed from pasta to water. The third reason may be the shape of the pasta. In our own study, six types of pasta were used, which are short and have many recesses, which means that their surface is large, larger than pasta with a smooth surface, such as spaghetti. The research showed that the shape and type of pasta significantly (*p* < 0.05) influenced the level of minerals in pasta, also when cooked (Figure 1 and Figure 2). Accordingly, the shape of the pasta can also influence the size of the retention factor. This may be explained by the fact that in the studies by Ranhotra et al. [22] and Albrecht et al. [23], the results were lower than those found in our own study, because the cited authors used other types of pasta—long, with regular shapes (spaghetti, noodles, macaroni).

### 4.2. Pasta as a Source of Minerals in the Diet

Cereal products are an important source of minerals in the diet of Poles. Their share in covering the requirement for respective elements can be presented as: Mn > Cu > Na > Fe > Zn > Mg > K > Ca [2,3,4]. Considering a variety of cereal products (bread, groats, rice, and cereal flakes), the maximum share of pasta in the supply of the analysed minerals is 5% [3,4,39]. Poles do not eat a lot of pasta—on average, 0.4 kg per capita [1]. Still, in Poland—like in many countries—pasta is the second most often consumed cereal food [1,39]. However, based on the results of our own study, it can be concluded that pasta is an important source of Mg, Cu, and Mn in the diet of Poles. At the same time, it was observed that the best source of minerals was pasta cooked in unsalted water without rinsing, so this is the best method of culinary treatment of pasta. Children receive the highest amount of Mg and Cu with one serving of pasta—above 26% RDA and above 12% RDA, respectively. On the other hand, pasta is the best source of Mn for adults and adolescents—it supplies 18–20% AI. Cubadda et al. [31] estimated that one serving of pasta available on the Italian market, weighing 80 g before cooking, would cover the daily requirement for minerals (RDA) of an adult as follows: 18% Cu, 10–14% Zn (for men and women, respectively), 5–9% Fe (for women and men, respectively). Ranhotra et al. [22] found that pasta in the diet of adult Americans is a good source of Mg, Zn, Fe, Cu, and Mn by comparing the measured values against American nutritional guidelines.

The diets of Poles are deficient in most minerals. Most frequently, it is noted that the diets of children, adolescents, and the elderly contain insufficient levels of Cu and Mg [40,41,42]. In addition, it was demonstrated that in women’s breastmilk, the content of Cu was too low, due to a deficiency of this element in their diet [43]. The deficiency of Mn in the human diet around the world is very rare, while an excessive intake of Mn with food can be a problem. The toxicity of Mn results primarily from its prooxidative effect, leading to damage of the cells of the extrapyramidal system, in particular of the dopamine system, which can lead to the development of Parkinson’s disease [44,45]. The identified toxic level of Mn is 9–11 mg for adults and 2–6 mg for children [46]. The diet of Poles contains excessive amounts of Mn; according to some data, the intake of Mn is twice the requirement for this element [38,47]. Our own study showed that rinsing pasta after cooking decreases the content of Mn in it. Although some sources mention an excessive intake of manganese from food, in our own study, the upper safety limit was not exceeded, even for non-rinsed and unsalted pasta.

Although our own studies do not imply a significant share of pasta in the supply of Na (less than 1% AI for all analysed age groups), a significant effect of adding salt to water while cooking on the supply of Na can be observed. The results clearly indicate that the least Na will be supplied with pasta cooked in unsalted water. This is particularly important, since the intake of Na as table salt in the diet of Poles is excessive, and every method to reduce the consumption of Na is important, all the more so because correct nutritional habits are of key importance. Studies involving people with hypertension showed that although the respondents were aware of the recommendations to reduce the consumption of salt, most of them exceeded the recommended levels [48]. Children and adolescents also consume too much salt—on average, more than 3 g a day [49,50]. As early as 2006, the World Health Organization recommended that the daily intake of Na should not exceed 2 g, which is an amount contained in 5 g of table salt, to reduce the risk of cardiovascular diseases [51]. In Western societies, the consumption of salt ranges from 8 to 12 g, which is several times higher than the physiological requirement of the body amounting to about 0.5 g [52]. In Poland, two times the recommended norm of salt is consumed mainly as table salt added to meals [29]. A tendency to consume excessive amounts of salt develops at an early stage of life, when adults—guided by their own taste—use excessive amounts of salt when preparing food for children [53]. The intake of Na can be decreased by gradually decreasing the content of this element in the most popular foodstuffs or changing the culinary treatment method, that is, adding salt after cooking, and not using salt in cooking, such as for pasta or groats. Studies in Peru showed that as many as 84% of people preparing meals at home were willing to reduce the use of salt during culinary processes [54].

## 5. Conclusions

The most efficient way of reducing the intake of Na and increasing that of deficient minerals with cereal products is not adding table salt to water in which pasta is cooked, even if the producer recommends adding salt to water. Only additions to pasta, such as sauces, should be seasoned, whilst one should reduce the amount of salt when seasoning, and instead of salt, use herbs and natural spices for flavour. American studies showed that the diet of people eating pasta is richer in deficit nutrients, including minerals, but on the condition that the use of table salt and saturated fatty acids is reduced in cooking [55]. Because Na in the form of table salt is very soluble in water and easily diffused, the recommendation not to add salt to water when cooking may also apply to other foodstuffs, such as groats, rice, vegetables, and beans. This problem should be examined in detail. In turn, rinsing pasta after cooking leads to the flushing of minerals and decreases the nutritional value of pasta, so this culinary treatment should be discouraged. The right culinary processing of food can reduce the risk of diet-related diseases of civilisation. There is a need for more research regarding the relationship between the type of cooked pasta and the content of minerals.

## Figures and Tables

**Figure 1 foods-10-02131-f001:**
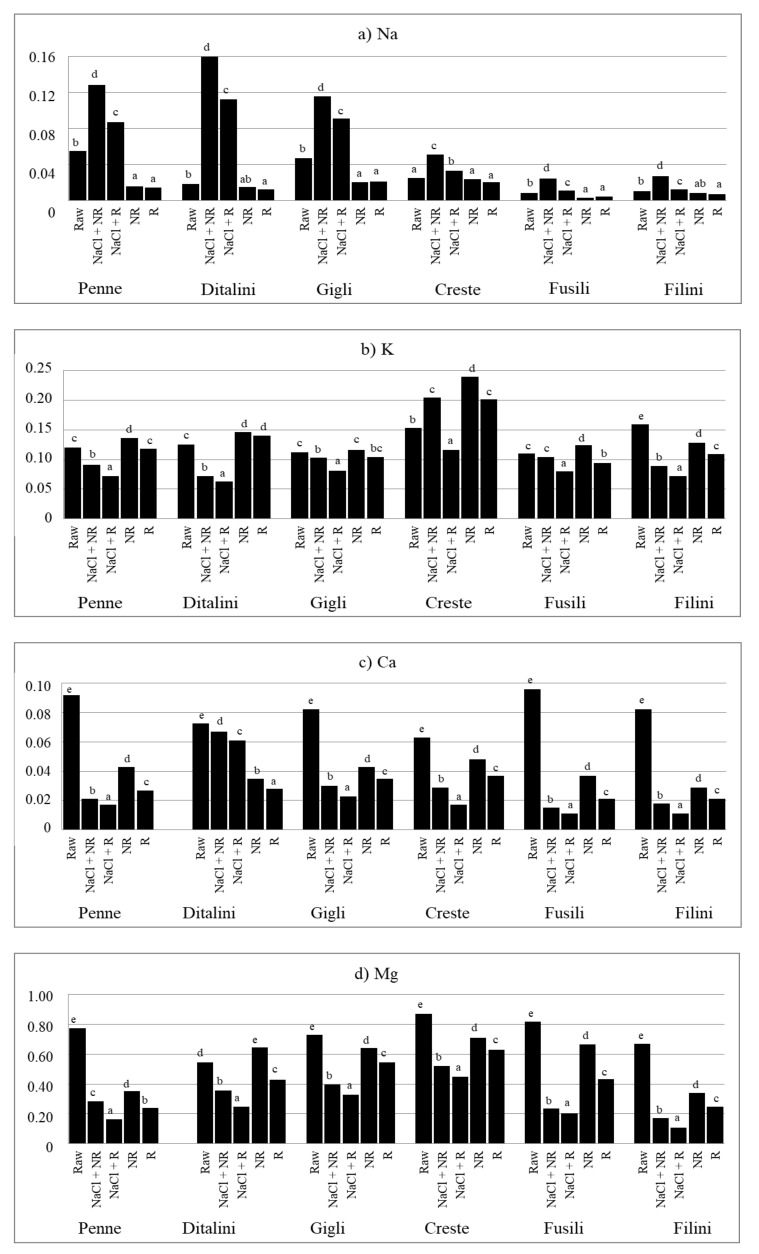
The content of: (**a**) Na; (**b**) K; (**c**) Ca; (**d**) Mg in pasta, g kg^−1^ fresh weight. Raw—raw pasta; NaCl + NR—cooked with salt and not rinsed; NaCl + R—cooked with salt and rinsed; NR—cooked without salt and not rinsed; R—cooked without salt and rinsed; ^a,b,c,d,e^—values with different superscripts differ at *p* < 0.05 by Duncan’s test.

**Figure 2 foods-10-02131-f002:**
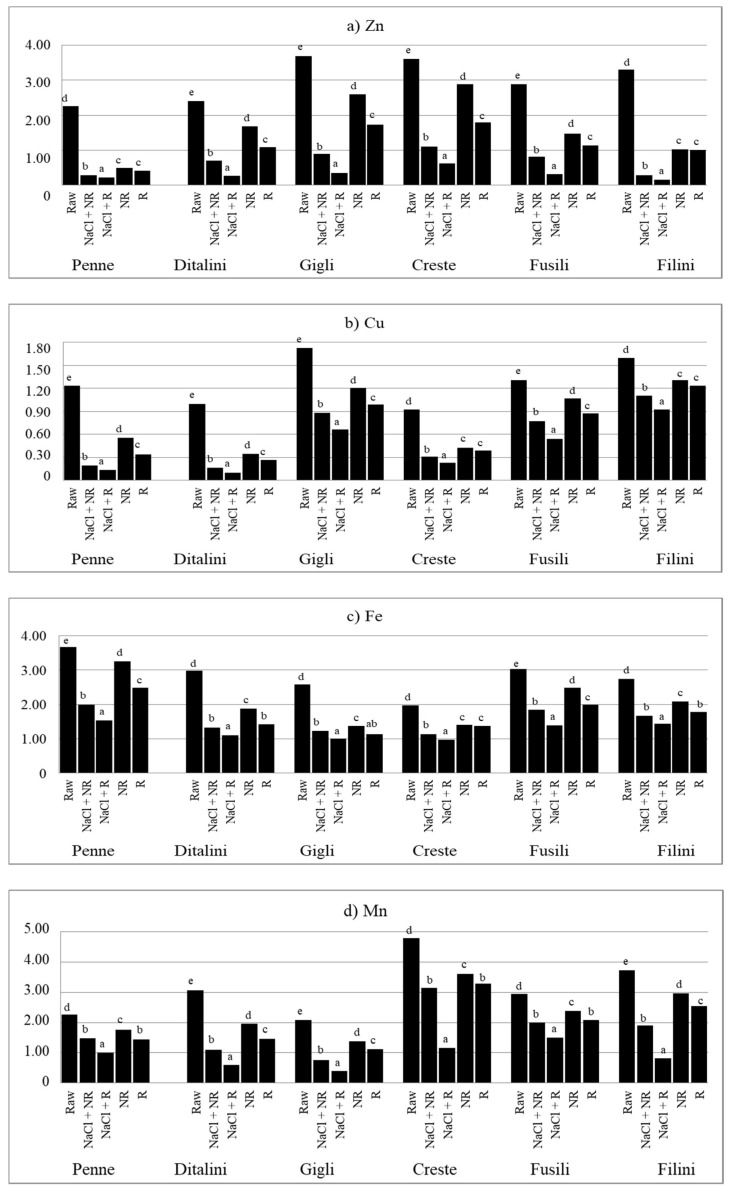
The content of: (**a**) Zn; (**b**) Cu; (**c**) Fe; (**d**) Mn in pasta, mg kg^−1^ fresh weight. Raw—raw pasta; NaCl + NR—cooked with salt and not rinsed; NaCl + R—cooked with salt and rinsed; NR—cooked without salt and not rinsed; R—cooked without salt and rinsed; ^a,b,c,d,e^—values with different superscripts differ at *p* < 0.05 by Duncan’s test.

**Figure 3 foods-10-02131-f003:**
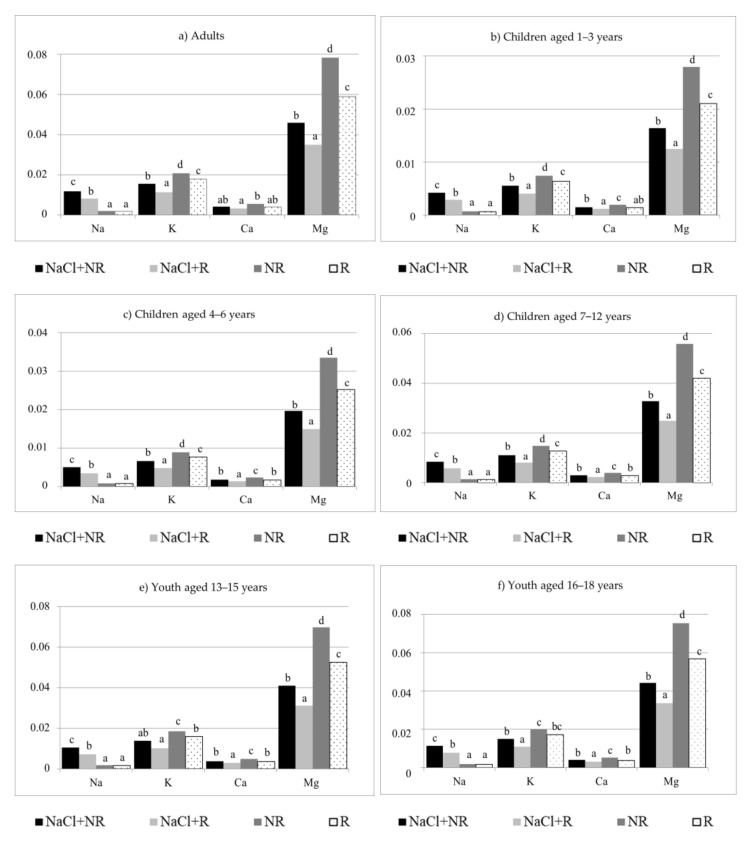
Intake of Na, K, Ca, and Mg with analysed pasta per portion, cooked, g. Portion size: (**a**) Adults 140 g [24], (**b**) children aged 1–3 years 50 g [25], (**c**) children aged 4–6 years 60 g [26], (**d**) children aged 7–12 years 100 g [27], (**e**) Youth aged 13–15 years 125 g [28], (**f**) Youth aged 16–18 years 135 g [28]; NaCl + NR—cooked with salt and not rinsed; NaCl + R—cooked with salt and rinsed; NR—cooked without salt and not rinsed; R—cooked without salt and rinsed; ^a,b,c,d^—values with different superscripts differ at *p* < 0.05 by Duncan’s test.

**Figure 4 foods-10-02131-f004:**
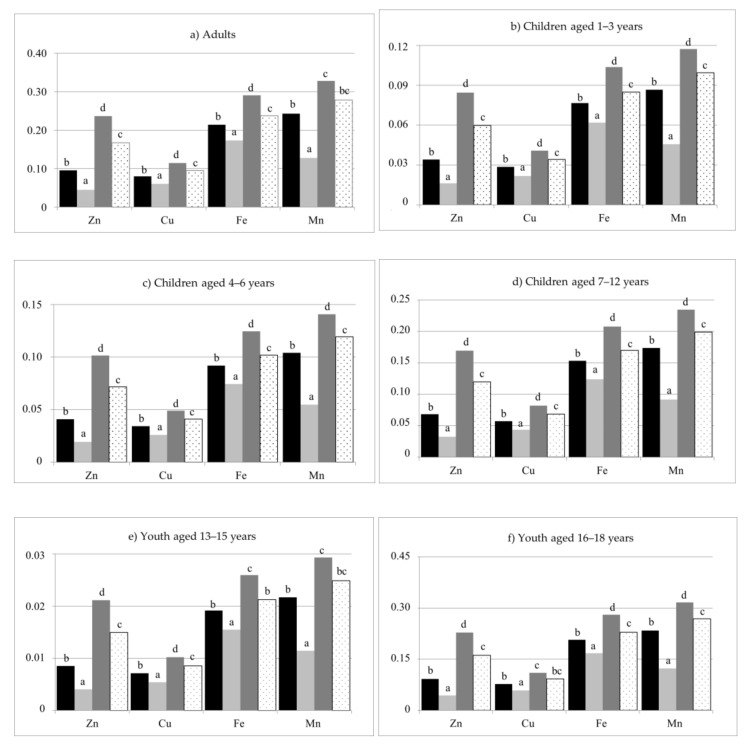
Intake of Zn, Cu, Fe, and Mn with analysed pasta per portion, cooked, mg. Portion size: (**a**) Adults 140 g [24], (**b**) children aged 1–3 years 50 g [25], (**c**) children aged 4–6 years 60 g [26], (**d**) children aged 7–12 years 100 g [27], (**e**) children aged 13–15 years 125 g [28], (**f**) children aged 16–18 years 135 g [28]; NaCl + NR—cooked with salt and not rinsed; NaCl + R—cooked with salt and rinsed; NR—cooked without salt and not rinsed; R—cooked without salt and rinsed; ^a,b,c,d^—values with different superscripts differ at *p* < 0.05 by Duncan’s test.

**Table 1 foods-10-02131-t001:** Characteristics of analysed products.

Pasta Type	Number of Samples	Cooking Time, min. (al dente)	Salt Amount in Raw Product, g per kg ^1^	Weight of Pasta before Cooking, g	Weight of Pasta after Cooking, g
Penne	7	7–13	0–0.3	100	294–320
Ditalini	5	5–12	0–0.05	100	290–311
Gigli	5	5–8	0–0.5	100	250–274
Creste	4	5–12	0–0.07	100	300–314
Fusilli	7	4–9	0–0.5	100	313–321
Filini	7	3–8	0–0.09	100	251–271

^1^ Values declared by producer.

**Table 2 foods-10-02131-t002:** Data of triplicate certified reference materials analysis.

	Na	K	Ca	Mg	Zn	Cu	Fe	Mn
Certified reference material	LGC-7173	LGC-7173	LGC-7173	LGC-7173	LGC-7173	LGC-7173	LGC-7173	LGC-7173
Certified, mg kg^−1^	1180	7480	17800	2037	78.00	14.00	148.0	90.00
Observed, mg kg^−1^	1263	7496	17523	2049	81.46	13.78	139.2	87.15
Recovery rate,%	107	100	98	101	104	98	94	97
Certified reference material	NCS ZC 73009	NCS ZC 73009	NCS ZC 73009	NCS ZC 73009	NCS ZC 73009	NCS ZC 73009	NCS ZC 73009	NCS ZC 73009
Certified, mg kg^−1^	17.00	1400	340.0	450.0	11.60	2.710	18.50	5.400
Observed, mg kg^−1^	15.91	1389	327.2	455.7	11.51	2.715	18.65	5.287
Recovery rate,%	94	99	96	101	99	100	101	98

**Table 3 foods-10-02131-t003:** The retention of Na, K, Ca, Mg, Zn, Cu, Fe, and Mn in products cooked with salt or without salt, and rinsed or not rinsed, % (percentage of mineral retention) *.

	Cooking Parameters				Influence of	
	NaCl + NR	NaCl + R	NR	R	Salt Addition	Rinsing after Cooking
					ANOVA *p* Values	
Na	780 ± 129 ^d^	308 ± 293 ^c^	118 ± 77 ^a^	269 ± 110 ^b^	0.002	0.001
K	254 ± 90 ^b^	187 ± 41 ^a^	341 ± 84 ^d^	293 ± 71 ^c^	0.018	0.018
Ca	116 ± 85 ^c^	90 ± 82 ^a^	148 ± 44 ^d^	107 ± 40 ^b^	0.004	0.002
Mg	134 ± 49 ^b^	100 ± 43 ^a^	231 ± 80 ^d^	177 ± 58 ^c^	<0.001	0.004
Zn	67 ± 29 ^b^	32 ± 13 ^a^	160 ± 68 ^d^	114 ± 35 ^c^	<0.001	0.012
Cu	121 ± 66 ^b^	91 ± 55 ^a^	178 ± 62 ^d^	148 ± 63 ^c^	0.003	0.010
Fe	163 ± 28 ^b^	132 ± 22 ^a^	218 ± 51 ^d^	180 ± 42 ^c^	0.021	0.003
Mn	163 ± 53 ^b^	92 ± 48 ^a^	222 ± 33 ^d^	186 ± 37 ^c^	0.011	0.028

* Average values for samples, each in 3 replications; NaCl + NR—cooked with salt and not rinsed; NaCl + R—cooked with salt and rinsed; NR—cooked without salt and not rinsed; R—cooked without salt and rinsed; ^a,b,c,d^—values with different superscripts in the same rows differ at *p* < 0.05 by Duncan’s test.

**Table 4 foods-10-02131-t004:** Comparison of the mean content of Na, K, Ca, Mg, Zn, Cu, Fe, and Mn in products cooked with or without salt, and rinsed or not rinsed, %.

	Na	K	Ca	Mg	Zn	Cu	Fe	Mn
R vs. NaCl + R ^1^	344	−36.8	−17.4	−40.7	−73.0	−36.5	−27.1	−54.1
NR vs. NaCl + NR ^2^	492	−25.4	−23.4	−41.3	−59.7	−30.2	−26.2	−26.1
NaCl + NR vs. NaCl + R ^3^	−31.3	−27.2	−22.4	−23.9	−52.7	−24.1	−19.1	−47.3
NR vs. R ^4^	−8.42	−14.0	−28.1	−24.7	−29.3	−16.6	−18.2	−15.2

^1^ The values for cooked without salt, rinsed (R) were assumed as 100%; ^2^ The values for cooked without salt, not rinsed (NR) were assumed as 100%; ^3^ The values for cooked with salt, not rinsed (NaCl + NR) were assumed as 100%; ^4^ The values for cooked without salt, not rinsed (NR) were assumed as 100%.

**Table 5 foods-10-02131-t005:** Share of pasta products in the supply of Na, K, Ca, and Mg in the diets of respective groups of Poles, g.

Mean Share of Pasta in Reference Daily Intake ^1^, %				
	Na	K	Ca	Mg
Adults ^2^				
NaCl + NR	0.8 (W, M)	0.5 (W, M)	0.4 (W) 0.3 (M)	14.6 (W) 11.2 (M)
NaCl + R	0.5 (W, M)	0.3 (W, M)	0.3 (W) 0.3 (M)	11.1 (W) 8.5 (M)
NR	0.1 (W, M)	0.6 (W, M)	0.5 (W) 0.4 (M)	24.8 (W) 19.0 (M)
R	0.1 (W, M)	0.5 (W, M)	0.4 (W) 0.3 (M)	18.7 (W) 14.4 (M)
Children, age 1–3 ^3^				
NaCl + NR	0.5 (W, M)	0.8 (W, M)	0.3 (W, M)	20.0 (W, M)
NaCl + R	0.4 (W, M)	0.5 (W, M)	0.1 (W, M)	15.0 (W, M)
NR	0.1 (W, M)	0.9 (W, M)	0.3 (W, M)	35.0 (W, M)
R	0.1 (W, M)	0.8 (W, M)	0.1 (W, M)	26.3 (W, M)
Children, age 4–6 ^4^				
NaCl + NR	0.5 (W, M)	0.6 (W, M)	0.2 (W, M)	15.4 (W, M)
NaCl + R	0.3 (W, M)	0.5 (W, M)	0.1 (W, M)	11.5 (W, M)
NR	0.1 (W, M)	0.8 (W, M)	0.2 (W, M)	26.2 (W, M)
R	0.1 (W, M)	0.7 (W, M)	0.2 (W, M)	19.2 (W, M)
Children, age 7–12 ^5^				
NaCl + NR	0.6 (W, M)	0.5 (W, M)	0.3 (W, M)	17.8 (W, M)
NaCl + R	0.5 (W, M)	0.4 (W, M)	0.2 (W, M)	13.5 (W, M)
NR	0.1 (W, M)	0.7 (W, M)	0.4 (W, M)	30.3 (W, M)
R	0.1 (W, M)	0.6 (W, M)	0.3 (W, M)	22.7 (W, M)
Youth, age 13–15 ^6^				
NaCl + NR	0.7 (W, M)	0.5 (W, M)	0.3 (W, M)	11.4 (W) 10.0 (M)
NaCl + R	0.5 (W, M)	0.3 (W, M)	0.2 (W, M)	8.6 (W) 7.6 (M)
NR	0.1 (W, M)	0.6 (W, M)	0.4 (W, M)	19.4 (W) 17.1 (M)
R	0.1 (W, M)	0.5 (W, M)	0.3 (W, M)	14.7 (W) 12.9 (M)
Youth, age 16–18 ^7^				
NaCl + NR	0.7 (W, M)	0.4 (W, M)	0.3 (W, M)	12.2 (W) 10.7 (M)
NaCl + R	0.5 (W, M)	0.3 (W, M)	0.2 (W, M)	9.4 (W) 8.3 (M)
NR	0.1 (W, M)	0.6 (W, M)	0.4 (W, M)	20.1 (W) 18.3 (M)
R	0.1 (W, M)	0.5 (W, M)	0.3 (W, M)	15.8 (W) 13.9(M)

^1^ Based on Jarosz [28]: RDA for Ca, and Mg; AI for Na, and K; Portion size: ^2^ Adults 140 g [24], ^3^ children aged 1–3 years 50 g [25], ^4^ children aged 4–6 years 60 g [26], ^5^ children aged 7–12 years 100 g [27], ^6^ youth aged 13–15 years 125 g [28], ^7^ youth aged 16–18 years 135 g [28]; NaCl + NR—cooked with salt and not rinsed; NaCl + R—cooked with salt and rinsed; NR—cooked without salt and not rinsed; R—cooked without salt and rinsed; W—women; M—men.

**Table 6 foods-10-02131-t006:** Share of pasta products in the supply of Zn, Cu, Fe, and Mn in the diets of respective groups of Poles.

Mean Share of Pasta in Reference Daily Intake ^1^, %				
	Zn	Cu	Fe	Mn
Adults ^2^				
NaCl + NR	1.2 (W) 0.9 (M)	8.9 (W, M)	1.2 (W) 2.6 (M)	13.5 (W) 10.6 (M)
NaCl + R	0.6 (W) 0.4 (M)	6.7 (W, M)	1.0 (W) 1.7 (M)	7.1 (W) 5.6 (M)
NR	3.0 (W) 2.2 (M)	12.7 (W, M)	1.6 (W) 2.9 (M)	18.3 (W) 14.3 (M)
R	2.1 (W) 1.5 (M)	10.7 (W, M)	1.3 (W) 2.4 (M)	15.4 (W) 12.1 (M)
Children, age 1–3 ^3^				
NaCl + NR	1.1 (W, M)	9.7 (W, M)	1.1 (W, M)	7.3 (W, M)
NaCl + R	0.5 (W, M)	7.3 (W, M)	0.9 (W, M)	3.8 (W, M)
NR	2.8 (W, M)	13.7 (W, M)	1.5 (W, M)	9.8 (W, M)
R	2.0 (W, M)	11.3 (W, M)	1.2 (W, M)	8.3 (W, M)
Children, age 4–6 ^4^				
NaCl + NR	0.8 (W, M)	8.5 (W, M)	0.9 (W, M)	6.9 (W, M)
NaCl + R	0.4 (W, M)	6.5 (W, M)	0.7 (W, M)	3.7 (W, M)
NR	2.0 (W, M)	12.3 (W, M)	1.3 (W, M)	9.4 (W, M)
R	1.4 (W, M)	10.3 (W, M)	1.0 (W, M)	7.9 (W, M)
Children, age 7–12 ^5^				
NaCl + NR	1.0 (W, M)	8.1 (W, M)	1.5 (W, M)	10.2 (W, M)
NaCl + R	0.5 (W, M)	6.1 (W, M)	1.2 (W, M)	5.4 (W, M)
NR	2.6 (W, M)	11.7 (W, M)	2.1 (W, M)	13.8 (W, M)
R	1.8 (W, M)	9.7 (W, M)	1.7 (W, M)	11.7 (W, M)
Youth, age 13–15 ^6^				
NaCl + NR	0.9 (W) 0.8 (M)	8.0 (W, M)	1.3 (W) 1.6 (M)	13.5 (W) 9.8 (M)
NaCl + R	0.4 (W) 0.4 (M)	6.0 (W, M)	1.0 (W) 1.3 (M)	7.1 (W) 5.0 (M)
NR	2.4 (W) 2.0 (M)	11.3 (W, M)	1.7 (W) 2.2 (M)	18.3 (W) 13.3 (M)
R	1.7 (W) 1.4 (M)	9.4 (W, M)	1.4 (W) 1.8 (M)	15.6 (W) 11.3 (M)
Youth, age 16–18 ^7^				
NaCl + NR	1.0 (W) 0.8 (M)	8.6 (W, M)	1.4 (W) 1.7 (M)	14.6 (W) 10.6 (M)
NaCl + R	0.5 (W) 0.4 (M)	6.6 (W, M)	1.1 (W) 1.4 (M)	7.8 (W) 5.6 (M)
NR	2.5 (W) 2.1 (M)	12.2 (W, M)	1.9 (W) 2.3 (M)	19.8 (W) 14.4 (M)
R	1.8 (W) 1.5 (M)	10.2 (W, M)	1.5 (W) 1.9 (M)	16.8 (W) 12.2 (M)

^1^ Based on Jarosz [28]: RDA for Ca, and Mg; AI for Na, and K; Portion size: ^2^ Adults 140 g [24], ^3^ children aged 1–3 years 50 g [25], ^4^ children aged 4–6 years 60 g [26], ^5^ children aged 7–12 years 100 g [27], ^6^ youth aged 13–15 years 125 g [28], ^7^ youth aged 16–18 years 135 g [28]; NaCl + NR—cooked with salt and not rinsed; NaCl + R—cooked with salt and rinsed; NR—cooked without salt and not rinsed; R—cooked without salt and rinsed; W—women; M—men.

**Table 7 foods-10-02131-t007:** Ranking of the content of minerals for each cooking method.

Cooking Parameters	
NaCl + NR	Mg > K > Na > Ca > Mn > Fe > Zn > Cu
NaCl + R	Mg > K > Na > Ca > Fe > Mn > Cu > Zn
NR	Mg > K > Ca > Na > Mn > Fe > Zn > Cu
R	Mg > K > Ca > Na > Mn > Fe > Zn > Cu

NaCl + NR—cooked with salt and not rinsed; NaCl + R—cooked with salt and rinsed; NR—cooked without salt and not rinsed; R—cooked without salt and rinsed.

## Data Availability

None.

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
