# Peer review of "Pasta as a Source of Minerals in the Diets of Poles; Effect of Culinary Processing of Pasta on the Content of Minerals"

_foods, 2021, doi:10.3390/foods10092131_

Round 1

Reviewer 1 Report

Review Comments

Title

 Pasta as a source of minerals in the diets of Poles; effect of culinary processing of pasta on the content of minerals

Abstract

Abstract is well written, provides a good summary of the research objectives, methods, and results gathered from the study.

Introduction

Introduction is also good as it provides enough background context in terms of the importance of pasta in Polish diet as well as the dietary habit problems concerned with the mineral content of cooked pasta as a result of conventional cooking processes.

Objectives are also well defined in the introduction part

Materials and Methods 

Materials and methods section also clearly describes the tests and sample preparation done for the study although the points below could be clarified in this section:

  • Was the replication for the mineral analysis taken from a single pasta prep or from three individual pasta prep samples? Authors only mentioned three replications for the chemical analyses (Line 107 -108)
  • Sample prep and analysis for the control sample (uncooked pasta) could be included. Was the mineral content of the control taken using the same analysis or based on other method/ information?
  • Line 117 – 125: References could be included for the wavelength used for analysis of the minerals (if based on an established reference)

Results and Discussion

The authors clearly presented the results of the study as well as its relevance to pasta as a staple for the diet of Poles.  The discussion in this section is clearly supported by the tables and figures presented in the manuscript. Discussion also covers relevant journals to the presented study

The following recommendations could be considered:

  • A summary of the relationships/ ranking of the mineral content (e.g., Mg > K > Na > Ca > Fe > Mn > Cu > Zn – line 188) for each culinary method testing could be included (as an addition to the existing tables or as an individual summary table)
  • For Table 5, significant figures (letters) could be added in addition to the P-values included
  • Tables for the established Polish RDA and AI values could be included (either in the manuscript or as a supplementary table) for better visualization of the discussion about Pasta as a mineral source in the diets of poles (section 3.2)
  • Discussion on the effect of pasta type on mineral content could also be included since the authors stated that it also had a significant effect.

Conclusions

Conclusion is well supported by the gathered data in the study. Authors could also include some possible recommendations for future studies based on the observations gathered.

Reviewer 2 Report

General remarks.

This study is presenting an impressive amount of data on the impact of various cooking procedures for a range of pasta products on the intake of minerals. Key findings for readers are in my view -

  • By not adding salt to water during cooking the intake of Na decreased strongly to levels representing a negligible 0.1% of AI-
  • A significant increase in the amounts of minerals available for intake can be realized by shifting from cooking with salt + rinsing after cooking to using no salt + no rinsing
  • The diets of Poles are deficient in most minerals. Especially diets of children, adolescents insufficient levels of Cu and Mg
  • Pasta products contribute especially to the intake of the minerals Mg, Cu and Mn –and to a low, not significant degree to intake of other minerals. –Shifting from cooking with salt+rinsing to cooking without salt and no rinsing results in a > 1,5 to – to almost 3 fold  fold increase of intake
  • To be checked:  Intake of Mn in Poland is significantly above recommended intake. BUT still well below levels that are considered as toxic (11mg/day) PLS Check! 
    • NOTE If this is not the case, then the general recommendation in the abstract is not valid and should be adapted

The impact and readability of the paper will considerably increase when the information is presented in a more compact way,

  • without duplications and all the less relevant information now presented
  • more emphasis on key findings such as the importance of of pasta + cooking without Na and rinsing for Cu and Mg intake 

I sincerely hope that a check that intake levels of Mn after cooking without Na and no rinsing will remain well below toxic levels. If otherwise,  the over-all recommendation becomes very weak

ABSTRACT

  • Line 22 If I have read the results, e.g. Table 9 and 10. My conclusion is:

When pasta was cooked in unsalted water, it contained less Na  and BUT MORE other minerals than pasta cooked in salted water

  • Add a sentence on the relevance of pasta intake (and the shift to cooking with no Na, no rinsing) , for Mg and Cu intake –

RESULTS

Table 8 – Question – NS is mentioned in the text below the table but is absent IN the table. Was this forgotten? Pls improve.  . 

Line 183-200  and Tables 3, 4, 6, 7

Many correlations are mentioned in detail, but I don’t see its relevance for the key messages of this paper.   They are just mentioned, but don’t play any further role in the . Please delete all these text and tables.  They may be placed in the Supplementary section if you wish to mention it 

Table 5. %??  explain in the heading of this table which %  is meant here

Lines 211-217 – Pls skip these lines

3.2. Pasta as a source of minerals in the diets of Poles

Fig 3. Intake of Na, K, Ca,  Mg with analysed pasta per portion, cooked, g?

I don’t understand the vertical axis.-If the intake is in grams (as indicated) we see that Mg is much higher than Na ??  Please improve/ explain.

The Figures 3 and 4 and the Tables 9 and 10 show the results in a clear way. This is sufficient for the paper The long text does not add any value  – it only duplicated the data presented well in the Figures and Tables and can best be skipped. I you prefer to show precise values in % this can be done in the Supplementary materials section – preferably as tables and not as text.

DISCUSSION

This section needs a major revision, for content and a better lay-out

 More emphasis should be given to the highlights and really important points – see also my remarks above. Also  the Mn story needs to be expanded AND improved (With Google Scholar I found already some key remarks about RDI / AI (for adults 1,8 – 2,3.  and toxic levels: below 11mg/day) no toxicity

The lay-out needs to be improved. Pls use 4.1. 4.2 sub-headings in italics for separating parts of the text

Question/ suggestion: why are  all the data from the literature included for K, Ca, Fe and Zn, where the contribution of pasta of intake is minimal in Poland? Why not focus on Na, Mg, Cu and Mn. Possibly only mention others – like Fe, Zn, when in other countries these minerals play a more important role, contrary to the situation n Poland. 

Round 2

Reviewer 2 Report

I thank the author for her clear replies. One major question remains for me - after het explanation of what % of retention means in Table 3. Many retention figures are well above 100%. Apart from Na, one would expect percentages below 100%. This question SHOULD deserve an explanation, for instance in the Discussion. Albrecht [23] reports retention values - apart from Na and Ca below or slightly above 100%, 

As shown in file:///C:/Users/kampjwvd/Downloads/39-45%20(1).pdf (2007) tap water in Lublin has high levels of minerals. This may explain  high retention values - for Mg en K far above 100% - in Table 3 (I did not make a detailed check) 

Further on, I still don't see the value of all the correlations between some of the minerals, since these correlations are not used or explained further on in the paper, but I respect the wish of the authors for keeping this type of  information in the paper. 
